# Parkinson’s Disease: Personalized Pathway of Care for Device-Aided Therapies (DAT) and the Role of Continuous Objective Monitoring (COM) Using Wearable Sensors

**DOI:** 10.3390/jpm11070680

**Published:** 2021-07-19

**Authors:** Vinod Metta, Lucia Batzu, Valentina Leta, Dhaval Trivedi, Aleksandra Powdleska, Kandadai Rukmini Mridula, Prashanth Kukle, Vinay Goyal, Rupam Borgohain, Guy Chung-Faye, K. Ray Chaudhuri

**Affiliations:** 1Department of Neurosciences, Institute of Psychiatry, Psychology & Neuroscience, King’s College London, London WC2R 2LS, UK; l.batzu@nhs.net (L.B.); Valentina.leta@nhs.net (V.L.); dhaval.trivedi1@nhs.net (D.T.); aleksandra.powdlewska@nhs.net (A.P.); Guy.chung-faye@nhs.net (G.C.-F.); Ray.chaudhuri@nhs.net (K.R.C.); 2Parkinson’s Foundation Centre of Excellence, King’s College Hospital, London SE5 9RS, UK; 3Nizams Institute of Medical Sciences, Hyderabad 500082, India; rukminimridula@gmail.com (K.R.M.); b_rupam@hotmail.com (R.B.); 4Vikram Hospitals, Benguluru 560052, India; drprashanth.lk@gmail.com; 5Medanta Institute of Neurosciences, New Delhi 122001, India; drvinaygoyal@gmail.com

**Keywords:** advanced Parkinson’s disease (APD), precision medicine, apomorphine subcutaneous infusion therapy, pain, intrajejunal, levodopa, motor and non-motor symptoms, PKG (KinetiGrap)

## Abstract

Parkinson’s disease (PD) is a chronic, progressive neurological disorder and the second most common neurodegenerative condition. Advanced PD is complicated by erratic gastric absorption, delayed gastric emptying in turn causing medication overload, and hence the emergence of motor and non-motor fluctuations and dyskinesia, which is initially predictable and then becomes unpredictable. As the patient progresses to the advanced stage, advanced Parkinson’s disease (APD) is characterized by refractory motor and non motor fluctuations, unpredictable OFF periods, and troublesome dyskinesias. The management of APD is a complex affair. There is growing recognition that GI dysfunction is common in PD, with virtually the entire GI system (the upper and lower GI tracts) causing problems from dribbling to defecation. The management of PD should focus on personalized care addressing both motor and non-motor symptoms, ideally including not only dopamine replacement but also associated non-dopaminergic circuits, particularly focusing on noradrenergic, serotonergic, and cholinergic therapies bypassing the gastrointestinal tract (GIT) by infusion or device-aided therapies (DAT), including levodopa–carbidopa intestinal gel infusion, apomorphine subcutaneous infusion, and deep brain stimulation, which are available in many countries for the management of the advanced stage of Parkinson’s disease (APD). The PKG (KinetiGrap) can be used as a continuous objective monitoring (COM) aid, as a screening tool to help to identify advanced PD (APD) patients suitable for DAT, and can thus improve clinical outcomes.

## 1. Introduction

Parkinson’s disease is the second most common neurodegenerative disease, affecting 1–2% of the population over the age of 60 [1,2]. Advanced Parkinson’s disease (APD) is associated with unmanageable, unpredictable motor and non-motor symptom fluctuations, which are refractory to standard oral/transdermal therapies, compromising quality of life (QOL) [3,4,5,6]. A recent consensus-based initiative based on a multi-country Delphi-panel (5-2-1) model, an approach to identifying functional indicators of advanced Parkinson’s disease, was externally validated in the OBSERVE-PD study [7]. This led to the development of the 5-2-1 motor paradigm (>5 oral levodopa doses/day, >2 h of “off” symptoms/day, and >1 h of troublesome dyskinesia/day [8]) used in clinical practice to identify advanced Parkinson’s patients and ensure timely referrals for device-aided treatments.

Managing advanced Parkinson’s disease is a complex affair. There is growing recognition that GI dysfunction is common in PD, with almost the entire GI system (the upper and lower GI tracts) causing problems from dribbling to defecation [9]. As the disease progresses, over 80% of patients with PD develop dysphagia and life-threatening aspiration pneumonia. Lower GI dysfunction results in slowed colonic transit, a reduced frequency of bowel movements, constipation, etc. [10,11,12,13]. The initial years with oral pulsatile dopaminergic treatment are relatively easy and effective. As patients reach the advanced stage, APD is complicated by erratic gastric absorption, delayed gastric emptying (causing medication overload), and poor levodopa absorption, hence the emergence of motor and non-motor fluctuations and dyskinesia, which is initially predictable and then becomes unpredictable [3,14,15,16]. Once unpredictable fluctuations or refractory “offs” start, one should start looking at non-oral infusion therapies or device-aided therapies (DAT) [17].

## 2. Available Infusion Therapies or Device-Aided Therapies (DAT) and Patient Selection 

In this situation, we should consider infusion or device-aided therapies, including levodopa–carbidopa intestinal gel infusion (LCIG), levodopa–entacapone–carbidopa intestinal gel infusion (LECIG), subcutaneous apomorphine infusion (APO), and deep brain stimulation (DBS). Many national guidelines have attempted to address the indications of these device-aided therapies (DAT) (Figure 1), and ideal patient selection remains somewhat of an unmet need [18].

If we look at the National Institute Centre of Excellence (Figure 2) indications for advanced device-aided therapies, the essential concept is based on offering the best medical therapy, which may start with subcutaneous apomorphine infusion (CSAI), or, if the symptoms are not adequately controlled, especially with severe dyskinesias, intrajejunal levodopa infusion (IJLI) or deep brain surgery (DBS) should be considered [19].

### 2.1. Selection of the Ideal Patient 

While therapeutic decisions and research on device-aided treatments have largely focused on the influence and effect on motor symptoms, we now know that Non Motor Symptoms (NMS)are an integral feature of PD and, therefore, should play a part (Figure 1) in the decision-making process for selecting the ideal patient [8,18].

### 2.2. Apomorphine: History and Molecular Structure

Apomorphine is considered one of the oldest antiparkinsonians. It is a drug found in water lilies that acts as an emetic, aphrodisiac, or hallucinogen [20]. In 1845, Adolf Edvard Arppe synthesized apomorphine from morphine and sulfuric acid [21]. In 1851, Thomas Anderson also synthesized apomorphine by heating codeine with sulfuric acid. It gained interest in medicine in 1868, when Matthiessen and Wright [22] heated morphine with concentrated hydrochloric acid and synthesized apomorphine hydrochloride. Figure 3 shows the history and evolution of apomorphine as a treatment for PD.

Apomorphine (C_17_H_17_NO_2_), a derivative of morphine, is a non-ergot dopamine agonist (DA) with high selectivity for D2, D3, D4, and D5 and, to a lesser extent, for D1 dopamine receptors. It activates serotonergic 5HT1A receptors but has antagonist effects on the serotonergic 5HT2A, 5HT2B, and 5HT2C receptors and adrenergic α2A, α2B, and α2C receptors [23].

While apomorphine has poor oral bioavailability (<4%), following its subcutaneous administration into the abdominal wall, 100% of it is rapidly absorbed. The time to peak plasma concentration is 10–60 min. Its concentration in the cerebrospinal fluid (CSF) peaks about 10–30 min later [23]. Its extremely lipophilic structure allows it to cross the blood–brain barrier. Its bioavailability after subcutaneous administration is similar to that after intravenous administration. It shows linear pharmacokinetics at 2–8 mg when a single subcutaneous injection is administered in the abdominal wall. Apomorphine is available in two presentations (Figure 4): A randomized double blinded study by Pfierffer et al. [24] looked at Continued efficacy and safety of subcutaneous apomorphine (Apo) in 62 patients with advanced Parkinson’s disease (APD) who had previously received APO for 3 months and placebo showed Significantly greater improvement in mean Unified PD rating scale motor scores in treatment group with no overall adverse event incidence observed in both groups supporting the the long-term use of intermittent APO as effective acute therapy for off episodes in advanced PD patients (APD) [24].

Compared with a placebo, apomorphine resulted in significantly and rapidly improved mobility, as assessed by an improvement in mean UPDRS motor scores, within a few minutes of administration. Maximal results were observed 20 min after administration. This effect persisted for at least 40 min after dosing [24]. In another study by Isaacson SH et al. [25], patients achieved an “on” state 37 min sooner, on average, with apomorphine injection than with oral levodopa, helping with early-morning akinesia, with a 61% reduction in the time to “on” (TTO) [25].

The Expert Consensus Group (Trenkwalder C et al. [26]) proposed the following clinical practice recommendations regarding the use of apomorphine in PD (Table 1 and Table 2).

Factors influencing supportive usage of APO.

Another recent multicentre, double-blind, randomized, placebo-controlled study (TOLEDO) [27] demonstrated the long-term efficacy of apomorphine infusion for motor fluctuations in PD. The significant reduction in off and increase in on time without troublesome side effects also led to substantial reductions in oral PD medication [27].

Levodopa–carbidopa (LD–CD) intrajejunal infusion (LCIG Figure 5) is a treatment in which traditional gold-standard levodopa in gel form is administered continuously into the primary site of levodopa absorption, the proximal jejunum. This is achieved via a percutaneous endoscopic gastrojejunostomy tube connected to a portable infusion pump. This was first launched in Sweden in 2004, after pioneering work by Professor Aquilonius and colleagues in Uppsala University, and it has now been on the market for 17 years [28].

An observational study (DUOGLOBE study) [29] evaluating the long-term (24 months follow up) effectiveness of LCIG in advanced PD (APD) patients, in which 20% of patients met all of the 5-2-1 criteria, showed sustained improvements in motor and non-motor scores and in quality of life (QoL), with supporting real-world data on the effectiveness, safety profile, and caregiver burden in APD patients. LCIG is probably the device-aided treatment for which we have the most robust evidence on the effect on NMS. In 2015, a systematic review identified eight open-label studies confirming that LCIG improved the NMS burden after a follow-up period ranging from 6 to 25 months, with specific positive effects on sleep and autonomic dysfunction, particularly gastrointestinal issues [30]. This was further explored and consolidated by the GLORIA registry, whose 24-month follow-up data showed a remarkable beneficial effect of LCIG on sleep disturbances, apathy, and gastrointestinal dysfunction [31].

DBS (Figure 6) is a widely accepted, conventional, and effective surgical treatment for Parkinson’s disease that involves implanting a device to stimulate targeted regions of the brain with electrical impulses generated by a battery-operated neurostimulator. DBS is thought to act by shifting the low-frequency (15–30 Hz) oscillatory activity observed in PD to a higher frequency, thus increasing the firing rate of the stimulated nucleus (commonly the sub-thalamic nucleus (STN), globus pallidus internus (GPi), or caudal zona incerta (cZi) [32]. Several randomized controlled trials (RCTs) comparing DBS (STN, GPi, or other) showed a superior efficacy and safety profile in patients with advanced Parkinson’s compared with basic medical dopaminergic treatment (BMT) [33,34].

### 2.3. DAT Therapies: Evidence-Based Clinical Motor and Non-Motor Outcomes

Parkinson’s disease (PD) is a progressive disorder. While the early motor phases of PD can be effectively managed by oral/transdermal dopaminergic therapy, as the disease progresses to advanced stages, it poses a challenge for neurologists to treat, complicated by the requirement to choose the ideal patients for device-aided therapies, including levodopa–carbidopa intestinal gel infusion (LCIG), levodopa–entacapone–carbidopa intestinal gel infusion (LECIG), subcutaneous apomorphine infusion (APO), and deep brain stimulation (DBS).

Personalizing treatment choices requires evidence and clinical-experience-based guidance for the device-aided management of PD, and it is paramount for better clinical outcomes. Several national guidelines and the Navigate PD program have attempted to address bespoke and ideal patient selection; the latter remains somewhat of an unmet need, as discussed above [18]. APO, LCIG, and bilateral STN-DBS have been available since early 2000 for the treatment of advanced Parkinson’s disease (APD) in many countries. Although several individual studies of LCIG, STN-DBS, and APO supported beneficial motor and non-motor outcomes [25,26,27,28,29,30,31], head-to-head comparative studies are limited. An open-label, non-randomized comparative study [35] (the Euroinf study) showed that, in advanced Parkinson’s patients, both IJLI and Apo infusion therapy appear to provide improvements in motor symptoms and quality of life, with IJLI resulting in better improvements in sleep/fatigue, gastrointestinal function, urinary domains, and sexual function compared to Apo [3].

Another prospective, multicentre, international, real-life cohort observation study of 173 PD patients, the Euroinf 2 study [36], the first and only study comparing all three device-aided treatments (APO, LCIG, and STN-DBS), was in agreement with previous studies [28,29,30,31,32,33,34,35,36]. It showed improvements in motor, non-motor, and quality-of-life outcomes [36]. However, interestingly, this study highlighted that each device-aided therapeutic option (DAT) showed biased outcomes in specific non-motor domains, with an overall reduction in non-motor burden. For instance, bilateral STN-DBS and LCIG appeared to benefit urogenital and gastrointestinal dysfunction, respectively, whereas APO showed supremacy in controlling attention/memory deficits. All three treatment options had a beneficial effect on depression and anxiety. Aspects of sleep dysfunction (insomnia, excessive daytime sleepiness, and restless leg syndrome) and fatigue improved with both LCIG and bilateral STN-DBS (Table 3 and Table 4), compared with APO, which showed a beneficial effect on perceptual problems and hallucinations. All three (STN-DBS, APO, and LCIG) had beneficial effects on the miscellaneous domain of the NMS scale, which incorporates unexplained pain, olfaction, weight changes, etc. Overall, this study highlights (Figure 7) the importance of personalizing therapeutic options based on holistic assessments of motor and non-motor symptoms [36].

Factors influencing supportive usage of IJLI.

Factors influencing supportive usage of STN-DBS.

### 2.4. Objective Measurements of Patient Outcomes in Parkinson’s Disease: Rating Scales

#### 2.4.1. MDS-UPDRS Scale 

This unified Parkinson’s disease rating scale (UPDRS) is a tool for monitoring the course of Parkinson’s and the degree of disability. The scale has three sections that evaluate key areas of disability, together with a fourth section that evaluates any complications of treatment [37,38].

Part I: Evaluation of mental activity, behaviour and mood, intellectual impairment, thought disorder motivation/initiative depression, sleep, pain, bladder and bowel problems, and fatigue. This subscale has scores from 0 to 4, with 4 representing the greatest level of dysfunction, and it can range from 0 (normal) to 16.

Part II: Self-evaluation of activities of daily living: speech, salivation, swallowing, handwriting, cutting food, dressing, hygiene, turning in bed, falling, freezing, walking, tremor, and sensory difficulties. This 14-item subscale ranges from 0 (normal) to 56.

Part III: Evaluation of motor function: speech, facial expression, tremor at rest, action tremor, rigidity, finger taps, hand movements, rotation of hands and forearms so palms face downward, rotation of hands and forearms so palms face upward, toe taps, leg agility, rising from chair, posture, gait, postural stability, and bradykinesia. This is the most commonly used subscale and has 14 different types of ratings, ranging from 0 to 4. The total score for subscale 3 ranges from 0 (normal) to 108, the sum of scores from 27 observations.

Part IV: Evaluation of complications of therapy; dyskinesia; early-morning “off” period deterioration, including the duration of “off” periods, predictability based on dosage, and whether onset is sudden or gradual; anorexia (including nausea and/or vomiting); and sleep disturbance. This subscale includes 11 questions, and the scores on this subscale range from 0 to 23.

#### 2.4.2. Hoehn and Yahr Rating Scale 

Hoehn and Yahr staging is probably the most widely known means for evaluating people with PD and was first described in 1967. It reflects motor manifestations of PD and is intended to reflect the degree of progression, combining features of motor impairment and disability, for scores of 0–5, with 0 = no signs of disease; 1 = unilateral disease (on one side); 1.5 = unilateral disease plus axial involvement; 2 = bilateral disease, without impairment of balance; 2.5 = bilateral disease, with recovery on the pull test; 3 = mild to moderate bilateral disease, needing assistance to prevent falling on the pull test, and physically independent; 4 = severe disability but still able to walk or stand unassisted; and 5 = wheelchair-bound or bedridden unless aided [39].

#### 2.4.3. Short Parkinson’s Evaluation Scale/Scales for Outcomes in Parkinson’s Disease (SPES/SCOPA) 

The SPES/SCOPA [40,41]. is a short, reliable, and valid scale used to evaluate the motor function of PD patients and includes three sections: A) Motor Evaluation (10 items, maximum of 42 points), B) Activities of Daily Living (7 items, 21 points), and C) Motor Complications (4 items, 12 points—with 2 items on motor fluctuations [6 points] and 2 on dyskinesias [6 points]). The response options for all the items range from 0 to 3.

#### 2.4.4. Non-Motor Symptoms Scale (NMSS) 

The Non-Motor Symptoms Scale (NMSS) [12] is a 30-item validated tool for assessing a wide range of non-motor symptoms in patients with Parkinson’s disease (PD). The NMSS measures the severity and frequency of a range of non-motor symptoms across nine dimensions: cardiovascular, sleep/fatigue mood/cognition, perceptual problems, attention/memory, gastrointestinal, urinary, sexual function, and miscellany. The score for each item is based on a multiple of severity (from 0 to 3) and frequency scores (from 1 to 4), for total scores of 0 (none) to 360.

#### 2.4.5. PDSS (Parkinson’s Disease Sleep Scale) 

The PDSS [42] is a simple bedside screening instrument for the evaluation of sleep disturbances in Parkinson’s disease. The PDSS is a visual analogue scale addressing 15 commonly reported symptoms associated with sleep disturbance. The 15 items are the overall quality of a night’s sleep (item 1), sleep onset and maintenance insomnia (items 2 and 3), nocturnal restlessness (items 4 and 5), nocturnal psychosis (items 6 and 7), nocturia (items 8 and 9), nocturnal motor symptoms (items 10–13), sleep refreshment (item 14), and daytime dozing (item 15). The severity of symptoms is reported by marking a cross along a 10 cm line (labelled from the worst to best state), and the scores for each item range from 0 (symptom severe and always experienced) to 10 (symptom-free). The maximum cumulative score for the PDSS is 150 (the patient is free of all symptoms).

#### 2.4.6. King’s Parkinson’s Pain Scale (KPSS) 

KPSS (King’s PD Pain Scale) [43] seems to be a reliable and valid scale for grading various types of pain in PD. Its seven domains (musculoskeletal pain, chronic pain, fluctuation-related pain, nocturnal pain, orofacial pain, discoloration/oedema/swelling, and radicular pain) include 14 items, with each item scored by severity (0–3) multiplied by frequency (0–4), resulting in a subscore of 0 to 12, with the total possible scores ranging from 0 to 168.

#### 2.4.7. Montreal Cognitive Assessment (MoCA) 

The Montreal Cognitive Assessment (MoCA) [44] is a widely used screening assessment for detecting cognitive impairment. It helps to assess several domains including memory recall, which involves two learning trials with five nouns, and delayed recall after approximately five minutes (scores out of 5 points), as well as visuospatial abilities using a clock drawing task (3 points) and a three-dimensional cube copy (1 point). Multiple aspects of executive function are assessed, by the trail-making B task (1 point), a phonemic fluency task (1 point), and a two-item verbal abstraction task (2 points).

Orientation to time and place is evaluated by asking the subject for the date on which and the city in which the test is occurring (6 points). Abstract reasoning is assessed (2 points). One point each is given for attention, concentration, and working memory, which are evaluated using a sustained attention task (target detection using tapping; 1 point), and digits forward and backward, as well as 3 points for a serial subtraction task. The assessment of language using three-item naming (familiar animals such as lions, camels, rhinos, etc.) scores 3 points, and repetition of two complex sentences scores 2 points.

The MoCA test is a one-page 30-point test, assessing several cognitive domains, and the MoCA scores range between 0 and 30. A score of 26 or over is considered to be normal; people with mild cognitive impairment (MCI) score an average of 22.1; people with Alzheimer’s disease score an average of 16.2.

#### 2.4.8. Hospital Anxiety and Depression Scale (HADS) 

HADS is a frequently used self-rating scale developed by Zigmond AS and Snaith RP for measuring anxiety and depression in non-psychiatric patients. The questionnaire comprises seven questions for anxiety (HADS Anxiety) and seven questions for depression (HADS Depression) [45]. The scoring for each item ranges from zero to three, with three denoting the highest level of anxiety or depression. A total subscale score of >8 points out of a possible 21 denotes considerable symptoms of anxiety or depression: 8–10 (mild), 11–14 (moderate), 15–21 (severe).

#### 2.4.9. Parkinson’s Disease Questionnaires (PDQ-8 and PDQ-39) 

The Parkinson’s Disease Questionnaire (PDQ-39) [46,47] is a validated disease-specific tool for measuring health-related quality of life in Parkinson’s disease patients. It covers eight dimensions—mobility, activities of daily of living, emotional well-being, stigma, social support, cognition, communication, and bodily discomfort—and it contains 39 questions. Each question is scored 0–4 points, transformed to a score ranging from 0 (good health) to 100 (poor health). The total score is derived from the sum of 39 scale scores divided by eight (the number of scales), which yields a score between 0 and 100 (100 = more health problems). This is equivalent to expressing the sum of all 39 item responses as a percentage score.

#### 2.4.10. Parkinson’s Disease Questionnaire (PDQ-8)

The PDQ-8 is a shorter questionnaire derived from the PDQ-39. It is an eight-question instrument with a question taken from each domain of mobility, activities of daily of living, emotional well-being, stigma, social support, cognition, communication, and bodily discomfort. The questions are scored 0–4, and the sum is taken.

## 3. Continuous Objective Monitoring (COM) Using Wearable Sensors and Its Role in Identifying Potential Candidates for Device-Aided Therapies (DAT)

After 5 years of disease [48,49], approximately 50% of PwP can develop motor fluctuations (bradykinetic fluctuations) and dyskinesia. Motor fluctuations and dyskinesia are the motor manifestations of reduced or excess (respectively) dopamine transmission, which also cause significant non-motor fluctuations [50]. Dyskinesias can sometimes be confused with tremor, and bradykinesia can be attributed to tiredness rather than a decline in the effectiveness of dopaminergic treatment. Some patients with cognitive issues have problems with compliance with their treatment, and in routine clinical practice, patient diaries are impractical and not commonly used apart from in clinical trials [51]. Objective measurement by capturing data during activities of daily living in the home environment helps not only with compliance but also with career burden, and for clinicians, it can provide continuous objective information that helps to optimize treatment and patient outcomes.

### 3.1. About PKG 

The Personal KinetiGraph^®^ (PKG^®^) Movement Recording System (Figure 8) is a new COM technology that provides scores for bradykinesia, dyskinesia, motor fluctuations, and tremor, as well as immobility as a proxy for daytime sleepiness. The Personal KinetiGraph (PKG) is a commercially available wrist-worn data logger system approved by the FDA, providing a continuous, objective, motor and ambulatory assessment of bradykinesia, dyskinesia, and motor fluctuations in PD. The logger is a smartwatch that is worn on the most affected wrist, weighs 35 g, and contains a rechargeable battery and a 3-axis iMEMS accelerometer. It provides data points every two minutes and produces a series of graphs and scores in a clinically useful format known as the PKG [52]. The device is water resistant. The logger is programmed to remind patients to take their PD medications by delivering vibrations, and consumption is acknowledged by swiping the logger’s smart screen. It also has sensors to detect whether the device is being worn.

The PKG is the graphical representation of the bradykinetic scores (BKS) and dyskinetic scores (DKS) collected every 2 min over an extended period of 6 days. It also provides sleep scores (as it is worn at night), daytime sleepiness scores, and inactivity [53], and also provides tremor scores [54]. The times at which medications are due and consumed are also shown, making it possible to assess whether there are dose-related variations in the BKS or DKS [55] (Figure 9).

The variables provided by the PKG are objective measures of these same factors that are considered clinically suitable candidates for DAT [18,56], which are recognized by the presence of increased “off” time and/or dyskinesia in subjects taking five or more doses/day [57]. Whilst there are many other factors taken into account before DAT is recommended, PKG is useful as a screening tool; for instance, the timing for deep brain stimulation (DBS) is important because there is a window of optimum benefit [58], and delay means that suitable candidates may have shorter benefit or lower benefit, or miss out on DBS entirely. Previous studies have shown [59,60] that 67% of patients referred for DBS are unsuitable for the procedure, yet only 1% of people with PD receive DBS [61], although as many as 20% may, in fact, be eligible [62].

One of the main reasons and indications for any DAT is motor fluctuations [63], which are frequently overlooked by both patients and clinicians [64]. The information from the PKG could be used to build a classifier (DAT classifier) that identifies patients eligible for DAT therapies with high sensitivity and specificity, correlating with the clinical criteria for DAT, and that can be used as a referral tool [65,66].

### 3.2. Glossary of PKG Terms 

The PKG produces a graphical representation of the BKS and DKS collected every 2 min over an extended period (typically 6 days) [52,53,54,67,68].

Median BKS. The median BKS was the 50th percentile of the BKS for all 6 days the PKG was worn (usually 6 days).The interquartile range of the BKS was a measure of the fluctuation of the BKS.The percent time in bradykinesia (PTB). Epochs whose BKS lay between 26.1 and 49.4 and whose 25th percentiles of the BKS were >18.5 and 90th percentiles, <80. Additionally, any epoch whose BKS was >49.9 but contained tremor was included.Median DKS: This is the 50th percentile for all the days that the PKG was worn. Brisk walking introducing resonant peaks may artificially increase the DKS. An algorithm was used to detect and remove epochs affected in this way.Interquartile range of DKS: calculates the median BKS and is a measure of the fluctuation of the DKS.Percent time in dyskinesia (PTD): Those DKS used to estimate the median DKS were passed through a median filter (most of the epochs in the filter period must be in the dyskinetic range (DKS > 7) for the centre to be classed as dyskinetic).Percent time with tremor (PTT): This was the percentage of 2 min epochs estimated over all the days that the PKG was worn that contained tremor. Tremor is likely to be present if the PTT score is >1%.The percent time immobile (PTI): This was the percentage of 2 min epochs with BKS > 80 from all the days that the PKG was worn. These scores were associated with daytime sleep.The doses of levodopa/day. These were calculated from the number of reminders programmed into the logger.

Bradykinesia was considered adequately treated if the BKS was <25, which relates to a Unified Parkinson’s Disease Rating (UPDRS) score of ~40 [52,53,54,67,68], and inadequately treated if the BKS was >25 [8,19,20,21,22,23]. Dyskinesia was considered “controlled” if DKS < 9, which relates to an Abnormal Involuntary Movement Score (AIMS) of 10 [52,53,54,67,68]. The percent time immobile (PTI) was defined as the percentage of 2-min periods between 9 AM and 6 PM where the movement data recorded by the PKG device were very low and correlated with the daytime sleep measured by polysomnography (PSG) and the Epworth Sleepiness Scale Scores (ESS). The percent time with tremor (PTT) was defined as the percentage of 2-min periods between 9 AM and 6 PM that contained tremor [68]. Tremor is likely to be present if the PTT score is >1% [52,53,54,67,68]. The other scores include compliance with the reminders.

### 3.3. PKG Database and Associated Studies

Currently we have a 6-year database (January 2012 to August 2018) with 27,834 complete and de-identified PKGs from 21 countries where the device has received regulatory approval. Data from seven countries (Australia, the UK, the USA, Sweden, Germany, the Netherlands, and France) where more than 500 PKGs had been performed (referred to as the Top 7 countries) were analysed, and these constituted 94% (26,112/27,834) of the PKGs in the database [52,53,54,67,68].

The first sub-analysis was based on the median scores of only those PD patients with serial PKGs (i.e., more than one PKG). There were statistically significant differences in BKS from the 1st to 2nd through to the 6th PKG readings in this stratified population (all *p* < 0.0001). The average time between each PKG order ranged from 23 to 42 days for the first 6 PKG readings. While the BKS improved by 3.3 points (30.9 to 27.6 points), the DKS increased by 0.3 points (0.8 to 1.1 points), suggesting improvements in the BKS due to clinicians optimizing the treatment regime [52,53,54,67,68]. Interestingly, these changes in treatment plan/dose optimization did not adversely affect the DKS, suggesting no significant increase in side effects or any abnormal movements.

## 4. Conclusions

PKG can be used as a COM in daily clinical practice. It aids in clinical decision making and the identification and quantification of PD motor symptoms, can be useful as a screening tool to help to identify advanced PD (APD) patients suitable for DAT, and improves clinical outcomes.

### 4.1. Clinical Scenario 1

A 64-year-old Asian patient (British Indian), a retired GP diagnosed with Parkinson’s disease 7 years ago, had an initial beneficial response to dopaminergic treatment and then presented with refractory motor (troublesome dyskinesias) and non-motor fluctuations (mild cognitive decline and non-intrusive perceptual issues, apathy, hallucinations, etc.). There were no obvious sleep-related issues or bowel/bladder complaints. Other problems included well-controlled type 2 diabetes treated with metformin monotherapy (1 g/day), and essential hypertension treated with captopril at 5 mg/day; there was no other significant past medical history, family history of dementia or history of allergies.

#### 4.1.1. Current PD Medications

Stalevo (l’dopa, 200 mg carbidopa, 50 mg; entacopone, 200 mg) QDS;Sinemet, controlled release, 250 mg (l’dopa, 200 mg; carbidopa, 50 mg) ON;Rotigotine, 8 mg (he responded very well initially and then started developing rashes, on rotigotine patches for 3 years);Previously tried a dopaminergic regime (selegiline, ropinorole, sinemet, etc.).

#### 4.1.2. Current Ongoing Problems

Troublesome dyskinesias;Unpredictable offs/freezing episodes;Attention/memory/cognitive problems;Apathy/hallucinations and non-intrusive perceptual issues.

### 4.2. Clinical Scenario 2

A 71-year-old Caucasian patient of Scottish heritage diagnosed with Parkinson’s disease 11 years ago, who had problems with dopamine agonists in the past (developed dopamine dysregulation syndrome with pramipexole and severe somnolence issues with ropinirole). They showed a good initial beneficial response to levodopa treatment, but then presented with unpredictable wearing offs, troublesome dyskinesias, and non-motor fluctuations, predominantly in terms of cardiovascular, urinary, and gastrointestinal dysfunction, as well as severe sleep-related issues (excessive daytime sleepiness). Other problems included symptoms suggestive of restless legs (RLS), with well-controlled hypertension treated with amlodipine at 5 mg/day, and no other significant past medical history.

#### 4.2.1. Current PD Medications

Sinemet PLUS (l’dopa, 100 mg; carbidopa, 25 mg) at 7 am, 10 am, 1 pm, 4 pm, and 7 pm;Sinemet, controlled release, 250 mg (l’dopa, 200 mg; carbidopa, 50 mg) at 10 pm;Opicopone, 50 mg, 8 pm;Previously tried a dopaminergic regime (pramipexole, ropinorole, and entacopone).

#### 4.2.2. Current Ongoing Problems

Troublesome dyskinesias;Unpredictable offs/freezing episodes/falls;Cardiovascular, urinary, and gastrointestinal dysfunction;Severe sleep-related issues (excessive daytime sleepiness);Previous adverse reactions to dopamine agonists.

### 4.3. Discussion and Outcomes

Patient 1. Being a medical practitioner who is well-versed about his condition and the available options, he is personally not keen on STN-DBS (patient preference). On the basis of the motor and non-motor profiles according to Euroinf 2 data, APO may represent a good therapeutic choice, keeping in line with the patient’s personal preference (not keen on surgery). He responded well to previous agonists (ropinorole/rotigotine). Based on the best medical therapy and available guidelines and evidence, APO (subcutaneous apomorphine infusion) was opted for, and the continuous, objective, motor, and ambulatory assessment of bradykinesia, dyskinesia, and motor fluctuations was performed to evaluate the efficacy of the device-aided therapy (apomorphine) with the wearable sensor monitor (COM) Personal KinetiGraph^®^ (PKG^®^).

Patient 2. Elderly gentleman with a history of previous adverse events in response to dopamine agonists (DDS) and with motor and non-motor (mainly cardiovascular, gastrointestinal, and sleep-related) problems and falls. On the basis of motor and non-motor profiles according to Euroinf 2 data, intrajejunal levodopa infusion may represent a good therapeutic choice, in keeping with the patient’s age and non-motor profiles. Surgery may not be a viable option, and due to a history of adverse events in response to dopamine agonists, APO is not indicated. Therefore, based on the best medical therapy and available guidelines and evidence, intrajejunal levodopa infusion (IJLI) was opted for, and the continuous, objective, motor, and ambulatory assessment of bradykinesia, dyskinesia, and motor fluctuations was performed to evaluate the efficacy of the device-aided therapy (IJLI) with the wearable sensor monitor (COM) Personal KinetiGraph^®^ (PKG^®^).

Overall clinical assessments revealed that both patients had refractory motor and non-motor fluctuations, unpredictable offs, and refractory freezing episodes, and both were on multi/varied dosing, with a combination of oral dopaminergic and transdermal dopamine treatments, with no obvious therapeutic effects or benefits compared to traditional conventional treatment. This indeed complements the Delphi model (5-2-1) [8] and was confirmed on COM (PKG recordings indeed showed variable BKS/DKS scores before the usage of DAT therapies, and Patient 1’s non-motor profile was dominated by mild cognitive decline, non-intrusive perceptual issues, apathy, hallucinations, etc.). Another factor to be considered for Patient 1 is how his personal preference was also implicated in the delivery of personalized advanced treatment. As he was not keen on surgery, according to available Euroinf 2 data, Apo (CSAI) [36] was considered the best option, and this was also the patient’s choice. He was monitored using COM (PKG), and 6-day recording showed an improvement in overall BKS/DKS scores (for the 20th to 14th percentiles before and after Apo (Figure 10, Figure 11 and Figure 12) respectively, and likewise for the bradykinesia scores).

Meanwhile, for our second patient, APO may not be suitable, as he has previously had problems with dopamine agonists, having developed dopamine dysregulation syndrome with pramipexole and severe somnolence issues with ropinirole. Other factors are also implicated, especially in this patient, in considering the delivery of personalized advanced treatment. His age, for instance, represents a key aspect in the assessment for DBS suitability; an age > 70 or 75 years is an exclusion criterion for DBS in many centres given the associated higher risk of complications as discussed previously [18,19]. Based on his current non-motor profile, LCIG was considered, as it showed superior efficacy in improving gastrointestinal, cardiovascular, and sleep-related problems and falls, and like our first patient, a 6-day PKG/COM recording was obtained (Figure 13) and showed an overall improvement in dyskinesias/fluctuating offs/bradykinesia scores.

Device-aided non-oral therapies are now considered and recommended worldwide for the management of advanced Parkinson’s disease. Personalizing the pathway of care and the successful delivery of these therapies depend on patient selection, motor and non-motor profiles, and patient choices and preferences. Body weight has also emerged as an important aspect in the decision-making process [69]. The PKG can be used as a COM in daily clinical practice, since it aids in clinical decision making and the identification and quantification of PD motor symptoms, is useful as a screening tool to help to identify advanced PD (APD) patients suitable for DAT, and improves clinical outcomes.

## Figures and Tables

**Figure 1 jpm-11-00680-f001:**
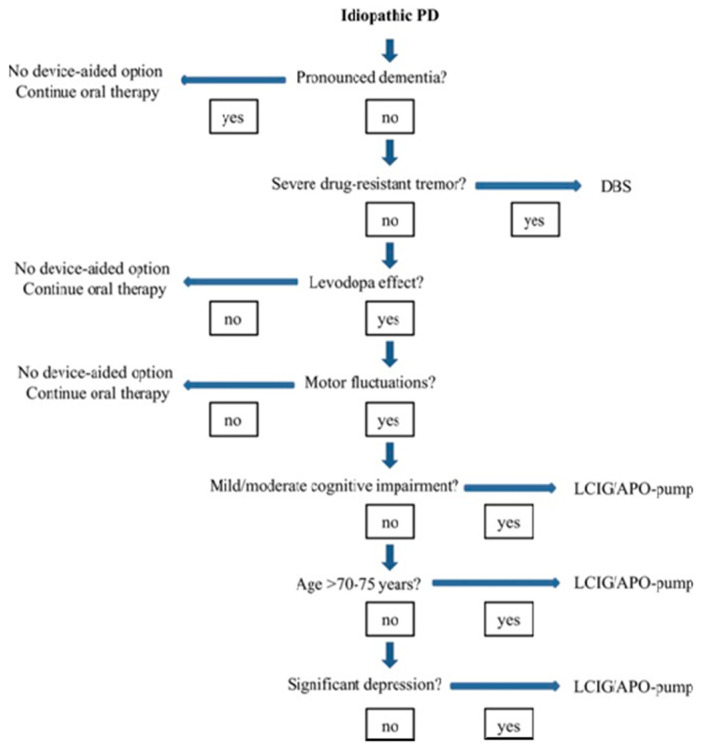
Patient selection for DAT therapies (Ref-Navigate PD) [18].

**Figure 2 jpm-11-00680-f002:**
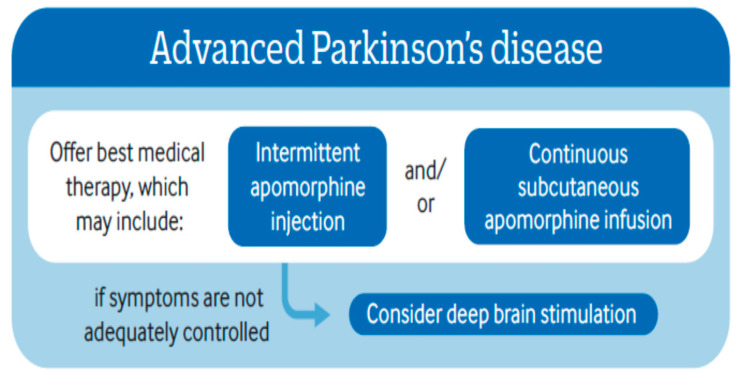
NICE guidelines 2017—Managing symptoms of Parkinson’s disease.

**Figure 3 jpm-11-00680-f003:**
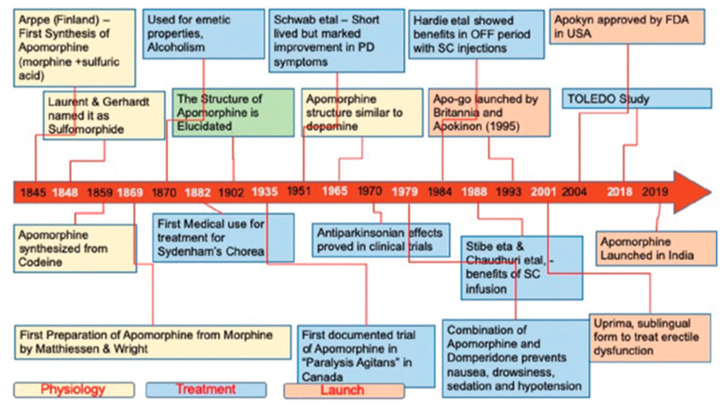
History and evolution of apomorphine as a treatment for PD.

**Figure 4 jpm-11-00680-f004:**
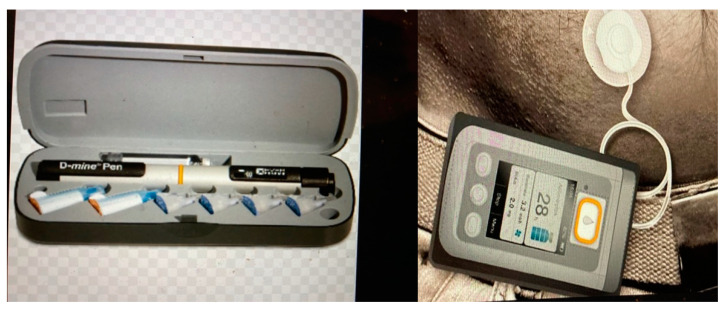
APO pen and pump.

**Figure 5 jpm-11-00680-f005:**
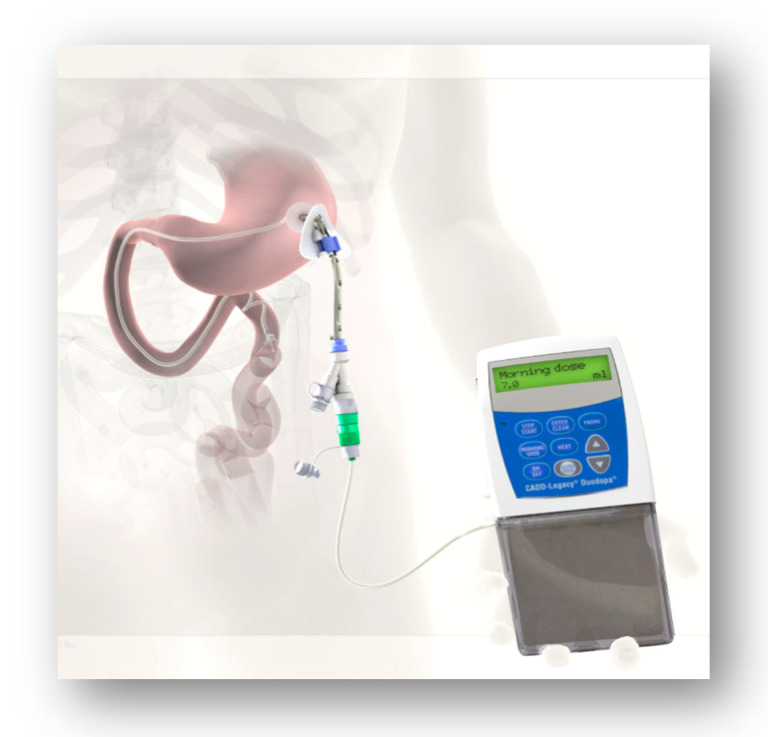
Intrajejunal levodopa infusion.

**Figure 6 jpm-11-00680-f006:**
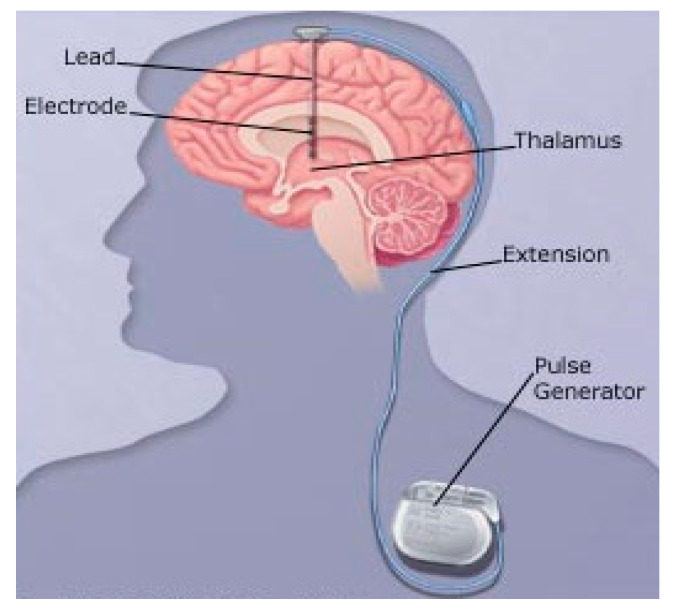
Deep brain surgery (DBS).

**Figure 7 jpm-11-00680-f007:**
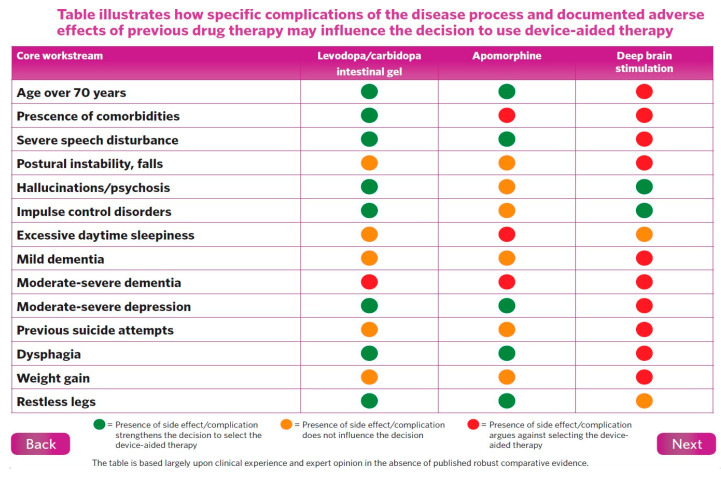
Euroinf 2 Study [36] showing DAT vs. specification of NMS domains for (DAT therapies—overall table).

**Figure 8 jpm-11-00680-f008:**
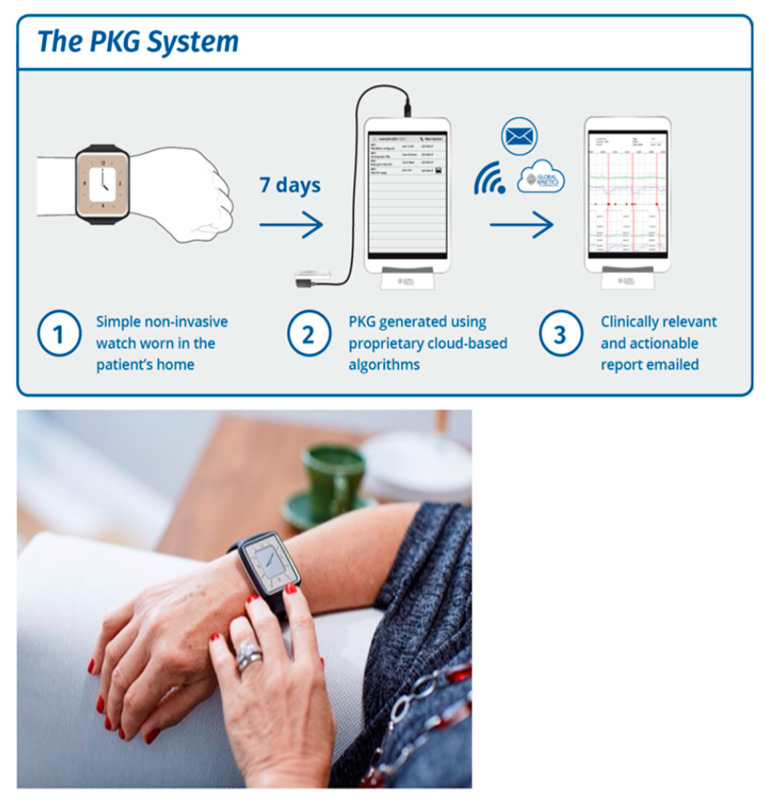
Monitoring Parkinson’s disease: PKG.

**Figure 9 jpm-11-00680-f009:**
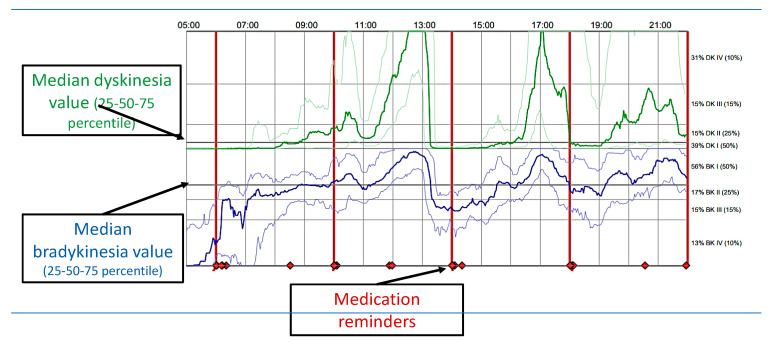
PKG: dyskinesia and bradykinesia.

**Figure 10 jpm-11-00680-f010:**
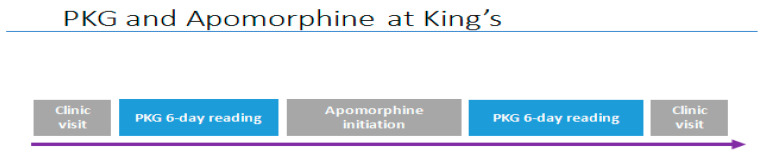
PKG and APO.

**Figure 11 jpm-11-00680-f011:**
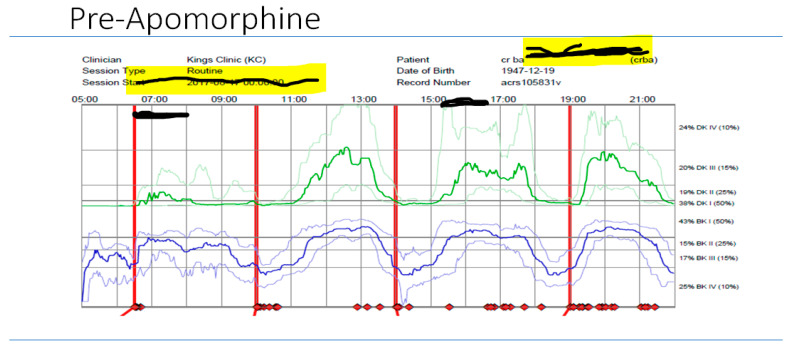
Pre APO.

**Figure 12 jpm-11-00680-f012:**
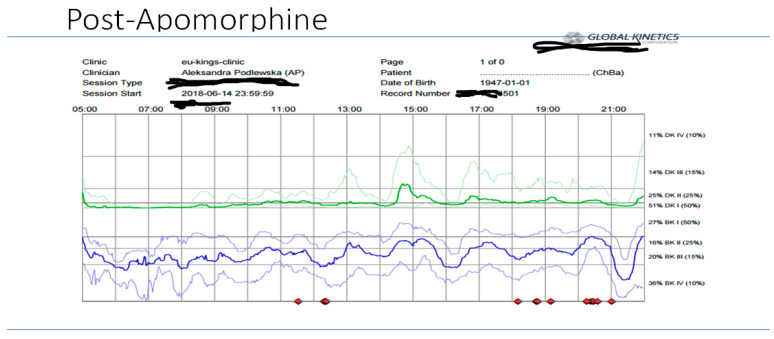
Post APO.

**Figure 13 jpm-11-00680-f013:**
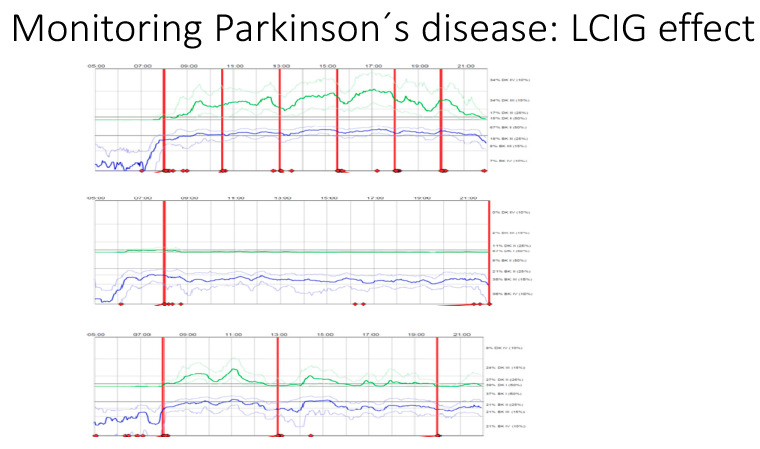
PRE and POST Duodopa.

**Table 1 jpm-11-00680-t001:** Expert Consensus Group report on the use of apomorphine in PD—clinical practice recommendations.

PEN (Figure 4)	PUMP (Figure 4)
Anticipated rescue when required during motor and non-motor “off” periods	Patient considers that rescue doses required too frequently
When absorption of oral levodopa is impaired or the patient has gastric emptying problems (gastroparesis)	Dyskinesias limit further therapy optimization
To treat delayed “on”	Simplify complex PD dosing regimens to improve convenience and compliance
To treat early-morning problems (akinesia and dystonia)	Alternative to surgical therapy or LCIG, if contraindicated, or due to patient preference
	Absorption or gastric emptying of oral levodopa is impaired

**Table 2 jpm-11-00680-t002:** Navigate PD: Factors influencing the use of CSAI.

Symptoms That Support Use	Symptoms That Discourage Use
Dyskinesias	Marked ongoing hallucinations/psychosis
Maintenance insomnia	Impulse-control disorders
Pronounced therapy-refractory depression	Drug-related daytime somnolence
Non-motor fluctuations	Orthostatic hypotension
Dysarthria	Marked ongoing hallucinations/psychosis
Restless legs	

**Table 3 jpm-11-00680-t003:** Navigate PD: Factors influencing the use of LCIG.

Symptoms That Support Use	Symptoms That Discourage Use
Dyskinesias	No specific symptoms (like severe dementia) to discourage use; presence of some symptoms may require further investigation
Drug-related hallucinations and/or delusions in patient history	
Impulse-control disorders	
Maintenance insomnia	
Mild cognitive impairment	
Pronounced therapy-refractory depression	
Dysarthria	
Restless legs	

**Table 4 jpm-11-00680-t004:** Navigate PD: Factors influencing the use of STN-DBS.

Symptoms That Support Use	Symptoms That Discourage Use
Dyskinesias	Marked ongoing hallucinations
Drug-related hallucinations and/or delusions in patient history	Dementia
Impulse-control disorders	Pronounced therapy-refractory depression
Maintenance insomnia	Dysphagia
Non-motor fluctuations	Dysarthria
	L-dopa-unresponsive postural and gait problems, falls
	Marked ongoing hallucinations

## Data Availability

Not applicable.

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
