# Peer review of "Parkinson’s Disease: Personalized Pathway of Care for Device-Aided Therapies (DAT) and the Role of Continuous Objective Monitoring (COM) Using Wearable Sensors"

_jpm, 2021, doi:10.3390/jpm11070680_

Round 1

Reviewer 1 Report

This review summarizes the body of evidence related to device aided therapies, continuous objective monitoring aid and health outcomes among patients with advanced PD. This can be valuable for providers caring for PD patients and facilitate identification of patients suitable for DAT use.  I have a few minor comments noted below: 

1. Page 6, Line 3: The sentence "Study by Pfeiffer et al (APO302)." did not make much sense to me. Please check and edit as appropriate.

2. Page 7 : The authors state, "An observational study evaluating the long effectiveness of LCIG in advanced PD (APD) patients who 20% of patients met all of the 5-2-1 criteria..". It authors add some detail on what long-term effectiveness means here?  

Author Response

Many thanks for taking time assessing my manuscript sir,

Reply to Reviewer #1:

We thank the reviewer for pointing this out,  we  deleted that phrase and explained in detail

Double blinded randomised control study by Pfeiffer et al and referenced it accordingly

Resultant changes:

Page 6, Line 3: The sentence "Study by Pfeiffer et al (APO302)." did not make much sense to me. Please check and edit as appropriate.

  • We elaborated the text as follows ( please also see attached/track changes in manuscript).

  • A randomised double blinded study by Pfierffer et al  looked at Continued efficacy and safety of subcutaneous apomorphine (Apo)  in 62 patients with advanced Parkinson's disease (APD who had previously received APO for 3 months and placebo showed Significantly greater improvement in mean Unified PD rating scale motor scores in treatment group with no overall adverse event incidence observed in both groups supporting the the long-term use of intermittent APO as effective acute therapy for off episodes in advanced PD patients.

Page 7 : The authors state, "An observational study evaluating the long effectiveness of LCIG in advanced PD (APD) patients who 20% of patients met all of the 5-2-1 criteria..". It authors add some detail on what long-term effectiveness means here?

  • We elaborated the text as follows ( please also see attached/track changes in manuscript).

  • 24 months outcome data showed sustained improvements of motor & non motor symptoms, also over all Quality of life and care giver burden with no obvious safety concerns in patients with Advanced Parkinson’s disease (APD).

Reviewer 2 Report

 Well written narrative review. Interest subject. Good reference list. Of note two minor  points:

  1. Checking referencing to  images and figures used and including them when absent
  2. Reformating text to one uniform style throuout the manuscript (e.g. "Current PD medication" section is different from the rest of the text) or summarising the text of interest in a schema (e.g. table)

Author Response

Reply to Reviewer #2 Comments :

  1. We thank the reviewer for  for taking the time to assess our manuscript and fort you kind comments.
  2. We thank the reviewer for pointing this out and actioned as per reviewer’s advice

Resultant changes:

  1. All Images/figures in entire Text ( total 13 figures and 4 tables) manually checked. Referenced wherever deemed necessary taking reviewer comment into consideration ( Pls see attached /highlighted track changes ).
  2. Entire manuscript carefully checked and re-editted with same uniform font type ( palantino linotype as per MDPI guidelines) including tables/figures referencing text and mainitained same font size through out ( please see attached /highlighted track changes)

Many thanks for taking time assessing my manuscript sir 

Regards
